# Reduced Toxicity of Trichothecenes, Isotrichodermol, and Deoxynivalenol, by Transgenic Expression of the *Tri101* 3-*O*-Acetyltransferase Gene in Cultured Mammalian FM3A Cells

**DOI:** 10.3390/toxins11110654

**Published:** 2019-11-10

**Authors:** Nozomu Tanaka, Ryo Takushima, Akira Tanaka, Ayaki Okada, Kosuke Matsui, Kazuyuki Maeda, Shunichi Aikawa, Makoto Kimura, Naoko Takahashi-Ando

**Affiliations:** 1Graduate School of Science and Engineering, Toyo University, 2100 Kujirai, Kawagoe, Saitama 350-8585, Japan; 2Graduate School of Engineering, Toyo University, 2100 Kujirai, Kawagoe, Saitama 350-8585, Japan; 3Department of Biological Mechanisms and Functions, Graduate School of Bioagricultural Sciences, Nagoya University, Furo-cho, Chikusa-ku, Nagoya, Aichi 464-8601, Japanmkimura@agr.nagoya-u.ac.jp (M.K.); 4Research Institute of Industrial Technology, Toyo University, 2100 Kujirai, Kawagoe, Saitama 350-8585, Japan

**Keywords:** trichothecene, biosynthetic pathway, acetyltransferase, deacetylase, deoxynivalenol, 3-acetyldeoxynivalenol, isotrichodermol, isotrichodermin

## Abstract

In trichothecene-producing fusaria, isotrichodermol (ITDol) is the first intermediate with a trichothecene skeleton. In the biosynthetic pathway of trichothecene, a 3-*O*-acetyltransferase, encoded by *Tri101*, acetylates ITDol to a less-toxic intermediate, isotrichodermin (ITD). Although trichothecene resistance has been conferred to microbes and plants transformed with *Tri101*, there are no reports of resistance in cultured mammalian cells. In this study, we found that a 3-*O*-acetyl group of trichothecenes is liable to hydrolysis by esterases in fetal bovine serum and FM3A cells. We transfected the cells with *Tri101* under the control of the MMTV-LTR promoter and obtained a cell line G3 with the highest level of C-3 acetylase activity. While the wild-type FM3A cells hardly grew in the medium containing 0.40 μM ITDol, many G3 cells survived at this concentration. The IC_50_ values of ITDol and ITD in G3 cells were 1.0 and 9.6 μM, respectively, which were higher than the values of 0.23 and 3.0 μM in the wild-type FM3A cells. A similar, but more modest, tendency was observed in deoxynivalenol and 3-acetyldeoxynivalenol. Our findings indicate that the expression of *Tri101* conferred trichothecene resistance in cultured mammalian cells.

## 1. Introduction

Trichothecenes are a group of mycotoxins produced by several filamentous fungi including *Fusarium*. They have a trichothecene skeleton of 12,13-epoxytrichothec-9-ene in common, but their side chain modification greatly varies, resulting in a large difference in their toxicity [1,2]. They exert toxicity mainly through the inhibition of protein synthesis in eukaryotes [3]. Some fusaria are notorious fungi, known to cause *Fusarium* head blight in important crops.

Based on the biosynthetic pathways, trichothecenes are largely divided into two groups: t-type trichothecenes, with a modifying group at C-3 position, and d-type trichothecenes, without a modifying group [4,5]. In the early biosynthetic pathway of trichothecenes, trichodiene is first synthesized through the cyclization of farnesyl pyrophosphate by Tri5p [6]. Trichodiene is then oxygenated to isotrichotriol, followed by spontaneous cyclization to produce the following t-type trichothecenes: *Fusarium graminearum* produces deoxynivalenol (DON), nivalenol, and their acetylated forms; *Fusarium sporotrichioides* produces T-2 toxin, neosalaniol, and diacetoxyscirpenol. On the other hand, in non-fusaria, including *Myrothecium*, *Trichoderma*, *Trichothecium, Stachybotrys,* and *Spicellum* [7], trichodiene is oxygenated to isotrichodiol followed by spontaneous cyclization to produce d-type trichothecenes. There is another classification method of dividing trichothecenes into four types (A–D) based on their chemical structures [8]. Type B group is distinguished from type A group by the presence of a ketone at C-8, and either type is synthesized by both t-type and d-type trichothecene producers. Type C group, which contains a 7,8-epoxide, and type D group, which contains a macrocyclic ring between C-4 and C-15, are exclusively produced by d-type trichothecene producers.

In fusaria, the first trichothecene produced is isotrichodermol (ITDol), which is immediately acetylated at the C-3 position by 3-*O*-acetyltransferase, Tri101p, into isotrichodermin (ITD) [5,9]. It has been reported that a 3-acetylated trichothecene is generally less toxic than its corresponding 3-hydroxy form [10,11,12,13]. Hence, it was suggested that Tri101p confers self-resistance to trichothecene produced in fusaria; indeed, *Schizosaccharomyces pombe* transformed with the *Tri101* gene has been found to be more resistant to T-2 toxin than the wild type (WT) [5]. So far, *Tri101* has been isolated from fusaria, including *F. graminearum* and *F. sporotrichioides*, and their enzyme kinetics have been evaluated extensively [14].

Trichothecenes are phytotoxins and may play a role as a virulence factor that contributes toxin producers to infect host plants [15,16,17]. Therefore, in order to combat *Fusarium* head blight, researchers have made extensive efforts to examine the effect of the transgenic expression of *Tri101* on trichothecene resistance and *Fusarium* infection in host cereals [18]. These studies have unequivocally proved that transgenic tobacco and rice showed improved trichothecene resistance [19,20]. In wheat, moderate tolerance to *Fusarium* infection was observed in a field trial [21]. On the contrary, in barley, deoxynivalenol may not be a virulence factor, and no effect on infection was observed in a field trial [22]. Thus, the gene manipulation of *Tri101* has been effective to confer tolerance to trichothecenes in some host plants, although the effect might be limited.

In contrast to the case with microbes and plants, the effects of trichothecene acetylation at the C-3 position and the transfection of *Tri101* into cultured mammalian cells are complicated to understand. Although there is approximately a 100-fold difference in terms of the in vitro inhibition of protein synthesis in rabbit reticulocytes between 3-acetylated trichothecenes and their corresponding 3-hydroxy forms, this difference was reduced to tenfold in terms of the in vivo inhibition of glycoprotein synthesis in BGK-21 cells [5]. Although it has been reported that a 3-acetyl trichothecene has lower toxicity than its corresponding 3-hydroxy form [2], there is a conflicting report of 3-acetyl T-2 toxin being as toxic as T-2 toxin in human cell cultures [23], and, so far, no reports have shown that improved resistance to trichothecenes is conferred by *Tri101* transfection in cultured mammalian cells. Moreover, it has been reported that the acetyl group at the C-3 position of trichothecene is easily removed in animal systems [24]. In this way, the toxicity of a C-3 acetyl trichothecene has tended to be inaccurately estimated by measuring the toxicity of a mixture of C-3 acetyl and C-3 hydroxy trichothecenes. Therefore, in this study, we attempted to maintain the 3-*O*-acetyl group attached to the trichothecene skeleton by the transgenic expression of *Tri101* in murine FM3A cells, resulting in a more accurate evaluation of the toxicity of 3-acetyl trichothecenes. We also examined whether *Tri101* transfection into mammalian cultured cells improves their resistance to 3-hydorxytrichothecenes.

## 2. Results and Discussion

### 2.1. Deacetylation of 3-Acetyldeoxynivalenol (3-ADON) and ITD

Assuming that the acetyl group of 3-acetyltrichothecenes was cleaved to produce more-toxic 3-hydroxytrichothecenes in cytotoxicity and animal studies, we first verified the extent of the deacetylation of two trichothecenes, 3-ADON, which is an acetylated form of the most common trichothecene DON, and, ITD, the first acetylated trichothecene produced by Tri101p in trichothecene biosynthesis.

First, 3-ADON or ITD was added to H_2_O, 125 mM Tris-HCl buffer (pH 6.5), and RPMI medium (without any additive), and the solutions were incubated in a CO_2_ incubator for 48 h. Neither DON nor ITDol was detected in H_2_O containing its corresponding 3-acetyltrichothecene, but some deacetylated forms were detected in both the buffer and the RPMI medium after incubation (Appendix A), which suggests that non-enzymatic deacetylation of these 3-acetylated trichothecenes occurred in them. Next, the actual culture medium for FM3A cells was examined. RPMI medium was supplemented with antibiotics, sodium pyruvate, β-mercaptoethanol, and deactivated fetal bovine serum (FBS). Considering the possibility that FBS contained contaminating esterases, we prepared the culture medium supplemented with non-boiled FBS (designated as N-medium) and boiled FBS (designated as B-medium). When 3-ADON or ITD was added to the media, the deacetylation rates of these trichothecenes were much higher in N-medium than B-medium, with ITD being deacetylated more efficiently than 3-ADON. While up to 96% of ITD was deacetylated to ITDol in the N-medium, only 4.6% of ITD was deacetylated in the B-medium. These results suggest that FBS contained broad-substrate-specificity esterases that could be deactivated by boiling, and ITD was more efficiently deacetylated than 3-ADON by these esterases.

FM3A cells were confirmed to grow in B-medium as normally as in N-medium; thus, the cells were incubated in B-medium containing 3-ADON or ITD in order to examine the stability of 3-*O*-acetyl group of these trichothecenes in a cell culture environment. As shown in Appendix A, 4.9% and 10.3% of 3-ADON was deacetylated to DON in 3 mL of cell culture medium containing 1 × 10^5^ cells and 6 × 10^5^ cells, respectively, while 17.0% and 39.1% ITD was deacetylated to ITDol, respectively. These results suggest that FM3A cells themselves contained esterases acting on C-3 of the trichothecene skeleton.

### 2.2. Transfection of FM3A Cells and Screening Cell Lines with High Expression of Tri101

First, we confirmed that the growth of FM3A WT cells was completely suppressed in the N-medium containing 30 µg/mL of blasticidin S and in the B-medium containing 0.1 µg/mL (0.40 µM) of ITDol. The growth of FM3A transfected with an empty vector was also completely suppressed in the B-medium containing 0.40 µM of ITDol. Thus, just after transfection of the vector carrying *Tri101*, the transformants were screened in N-medium with 60 µg/mL of blasticidin S and, subsequently, with increasing concentrations of the antibiotic (up to 250 µg/mL). Next, we transferred the screened cells into B-medium with 0.40 µM of ITDol, followed by B-medium with 0.80 µM of ITDol, in order to screen the transformants with a high Tri101p activity. Over 100 cell lines that survived the screening process were then cloned in B-medium containing 0.16 µM of ITDol by limiting dilution. Crude enzyme was prepared from each clone and the Tri101p activity toward ITDol or DON was initially evaluated by TLC, followed by HPLC. We selected one cell line, designated as G3, which showed the highest Tri101p activity. In the absence of the drugs, the growth rate of G3 was comparable to that of the WT cells.

### 2.3. Acetylase and Deacetylase Activities of Crude Cell Extracts from WT and G3 Cells

Next, we measured the in vitro acetylase activities of WT and the transformant G3 cells by HPLC. We added 169 µM DON or 200 µM ITDol at a final concentration to the reaction mixture, in order to measure the approximate V_max_, as these concentrations were much higher than the *K*_m_ values for DON and ITDol, 11.7 and 10.2 µM, respectively [14]. The crude enzymes from the WT cells showed no acetylase activities toward either substrate, with or without induction by dexamethasone (DEX; an inducer of transgene expression) (Figure 1, right). In contrast, the crude enzymes from the G3 cells showed acetylase activities toward both DON and ITDol. As expected, significantly higher acetylase activities were observed in the crude enzymes from G3 cells pretreated with DEX than in those without DEX (*p* < 0.05). This result is consistent with the previous observation that DEX resulted in a two- to fivefold increase in the levels of expression of a luciferase gene from the mouse mammary tumor virus-long terminal repeat (MMTV-LTR) promoter in FM3A cells [25]. All the crude enzyme prepared here showed deacetylase activities toward both 3-ADON and ITD (Figure 1, left).

### 2.4. Acquired Trichothecene Resistance in the Cells Transfected with Tri101

After the WT and G3 cells were seeded in the B-medium, the vehicle or ITDol was added to the culture medium, and the cell numbers were counted on day 3, 5, or 7 (Figure 2). Both WT and G3 cells grew normally until day 3 in the medium containing the vehicle, after which they underwent apoptosis. In contrast, in B-medium containing 0.40 µM ITDol, almost all the WT cells underwent apoptosis by day 3, but G3 cells continued to grow until day 5, after which the cell numbers decreased. In B-medium containing 0.80 µM ITDol, the G3 cells slowly continued to grow until day 5–7. In the medium containing a higher concentration of ITDol, even the G3 cells struggled to grow. Thus, it was concluded that G3 cells had acquired resistance to ITDol. This is the first study reporting that the transfection of *Tri101* into mammalian cells confers resistance to trichothecenes.

### 2.5. Cytotoxicity Evaluation of Each Trichothecene in WT and G3 Cells

We performed a water-soluble tetrazolium salts (WST) assay to evaluate the cytotoxicity of DON, 3-ADON, ITDol, and ITD more accurately, based on the assumption that the cytotoxicity of 3-acetyltrichothecenes had been overestimated in previous studies. First, the half maximal inhibitory concentration (IC_50_) values of ITDol and ITD in WT cells were determined using N-medium or B-medium (Appendix A). The observed IC_50_ values of ITDol and ITD were close when WT cells were incubated in N-medium (0.19 ± 0.01 and 0.26 ± 0.01 µM, respectively), however, a large difference was evident when assayed in B-medium (0.20 ± 0.01 and 2.14 ± 0.06 µM, respectively). The higher sensitivity of cultured mammalian cells to ITD in N-medium seems to be attributed to the contaminating esterases that act on ITD (Appendix A), resulting in the formation of much more toxic ITDol. This observation strongly suggests that without an elaboration to keep the 3-acetyl group attached to the ITDol ring, the toxicity of 3-acetyltrichothecene tends to be overestimated. Although the apparent IC_50_ value of 2.14 µM observed for ITD in B-medium could be closer to the actual IC_50_ value of intact ITD, this value may in fact be lower than the actual value, as an acetyl group may not remain attached to the C-3 position of the trichothecene skeleton due to the presence of intrinsic esterases hydrolyzing the 3-*O*-acetyl group in cultured mammalian cells (Figure 1, chemical reactions). Thus, we attempted to measure the concentration of ITDol possibly produced from ITD in the cell culture medium in this assay; however, ITDol was not detected in our HPLC analysis (limit of detection; 1.2 µM). As the IC_50_ value of FM3A cells against ITDol was as low as 0.22 µM (Figure 3), ITDol levels below the detection limit possibly affected cell growth. The higher toxicity of ITD may be explained by the C-3 deacetylation of ITD during the assay.

Next, in order to evaluate cytotoxicity of 3-ADON and ITD more accurately, we determined their IC_50_ values using G3 cells expressing *Tri101*. Figure 3A shows the dose–response cytotoxicity curves of DON and 3-ADON using WT and G3 cells. In WT cells, the difference in the IC_50_ values between DON and 3-ADON was around 12-fold. In G3 cells, all the IC_50_ values were increased significantly compared to their corresponding values in WT cells. The IC_50_ value in G3 cells pretreated with DEX was slightly higher than that without DEX, and the values were significantly different in DON, but not in 3-ADON. No significant difference was observed in the IC_50_ value of these trichothecenes in WT cells with and without DEX pretreatment. These results indicate that the transformant G3 cells were more resistant to both DON and 3-ADON than WT cells due to the expression of *Tri101* in G3 cells. As it is likely that the more accurate IC_50_ of 3-ADON was that of G3 cells treated with DEX (11.68 µM), this value was significantly higher than that of WT cells (7.09 µM without DEX, 6.45 µM with DEX) suggesting that cytotoxicity of 3-ADON might be overestimated in previous studies. Here, the IC_50_ of 3-ADON was at least 21-fold higher than that of DON (0.56 µM) obtained in WT.

The effect of the transgenic expression of *Tri101* in G3 was more obvious in the IC_50_ values of ITDol and ITD than in those of DON and 3-ADON. Figure 3B presents the result of the WST assay of ITDol and ITD in WT and G3 cells. In WT cells, the difference of the IC_50_ values between ITDol and ITD was 13–16 fold. In G3 cells, without DEX induction, the IC_50_ values of ITDol and ITD were almost doubled compared to those in the WT cells. Moreover, with DEX induction, the IC_50_ values of ITDol and ITD were three- to fourfold compared with those in the WT cells. This indicates that DEX induced *Tri101* expression, which resulted in the increased acetylation of ITDol to ITD. Thus, the IC_50_ values of both ITDol and ITD were almost doubled in G3 cells treated with DEX compared to those without DEX, and the values were significantly higher in both ITDol and ITD by DEX pretreatment. In WT cells, no significant increase was observed in the IC_50_ value of these trichothecenes by DEX pretreatment. Similar to DON and 3-ADON, a more accurate IC_50_ of ITD was likely to be that of G3 cells treated with DEX (9.58 µM), thus, this value was significantly higher than that of WT cells (2.78 µM without DEX, 3.59 µM with DEX), suggesting that cytotoxicity of ITD in previous studies might be overestimated. It represented an at least 44-fold increase of IC_50_ value of ITD (9.58 µM), which is supposedly to be more accurate, compared to that of ITDol obtained in WT (0.21–0.22 µM).

In this study, the level of *Tri101* expression was found to affect the IC_50_ values of trichothecenes in mammalian transformants. However, the *Tri101* expression level may not be sufficient to fully compete with the C-3 deacetylation activity of strong endogenous esterases. Nevertheless, this study clearly showed that the 3-acetyl group of 3-*O*-acetyltrichothecenes could be hydrolyzed enzymatically in mammalian cells and that their cytotoxicities were overestimated in other studies. The IC_50_ of trichothecenes were much lower than the *K_m_* of Tri101p against them, making it difficult to completely acetylate the C-3 of the trichothecenes added to the culture.

This is the first report to unambiguously demonstrate the acquired trichothecene resistance of cultured mammalian cells transformed with *Tri101*. These results strongly support the theory that C-3 acetylation blocks the toxicity of t-type trichothecenes, which serves as a self-defense mechanism of the producing organisms [5]. In contrast to fungi and plants, the strong C-3 deacetylase activity of mammalian cells activates 3-acetyltrichothecenes, the toxicity of which has been overestimated in previous studies.

## 3. Materials and Methods

### 3.1. Production and Purification of Trichothecenes

Each trichothecene was obtained from rice medium [26] or rice flour liquid medium [27] inoculated with *F. graminearum*: *Gibberella zeae* JCM 9873 for DON [28] and *F. graminearum* DSM 4528 for 3-acetyldeoxynivalenol (3-ADON) [29]. Isotrichodermin was obtained from the culture medium of *F. graminearum* MAFF 111233 *Fgtri11* disruptant (*Fgtri11^-^*, NBRC 113181) [27], while ITDol was obtained by the deacetylation of ITD in 2.8% ammonium solution. For purification, the ethyl acetate extract was applied to Purif-Rp2 equipped with Purif-Pack SI 30 µm SIZE (Shoko Scientific, Kanagawa, Japan), and the fraction containing a target trichothecene was concentrated. The concentrate was dissolved in ethanol and applied to preparative HPLC (UV detection at 254 nm for DON and 3-ADON, and at 195 nm for ITD and ITDol) equipped with a C_18_ column (Pegasil ODS SP100 10 φ × 250 mm, Senshu Scientific Co., Ltd., Tokyo, Japan). The concentration of each purified trichothecene was measured by HPLC equipped with a C_18_ column (Pegasil ODS SP100 4.6 φ × 250 mm) based on the standard curve previously obtained. The purity of each trichothecene used for the assays was confirmed to be >99%.

### 3.2. Maintenance of Cultured Cells

FM3A WT cells which were originally established from mammary carcinoma in C3H/He mouse were purchased from RIKEN BRC (Tsukuba, Japan). Cells were grown in RPMI1640 medium (Nacalai Tesque Co., Inc. Kyoto, Japan) with 10% FBS (Biowest, Nuaillé, France), penicillin/streptomycin (1000 units/mL each), 1 mM sodium pyruvate, and 50 µM 2-mercaptoethanol, denoted “N (non-boiled)-medium.” In order to deactivate any contaminating esterases in FBS, we prepared the same medium with FBS boiled at 100 °C for 5 min beforehand, denoted “B (boiled)-medium.” Cultured cells were incubated in a CO_2_ incubator (5% CO_2_, 37 °C).

### 3.3. Evaluation of Stability of 3-Acetyl Trichothecenes

Twenty micrograms of 3-ADON (59.1 nmol) or ITD (68.4 nmol) in ethanol was added into a 6 cm dish containing 3 mL of water, 125 mM Tris-HCl buffer (pH 6.8), and RPMI1640 medium without any supplements. Each solution was incubated in a CO_2_ incubator (5% CO_2_, 37 °C) for 48 h. Similarly, 3 mL of the N-medium and B-medium containing 10 µg of 3-ADON (29.6 nmol) or ITD (34.2 nmol) was incubated in a CO_2_ incubator for 48 h. We also prepared FM3A WT cells (3.3 × 10^4^/mL or 2 × 10^5^/mL) in 3 mL B-medium supplemented with 10 µg of 3-ADON or ITD. After 48 h, each solution or medium was extracted with the same volume of ethyl acetate twice. The extract was dried up under an N_2_ stream and resuspended in 200 µL ethanol. These concentrating steps were necessary for detection of the mycotoxins. The filtered samples were subjected to HPLC.

### 3.4. Plasmid Construction

*Tri101* was excised from pCold-His-TRI101 [30] by double digestion with *Nde*I and *Eco*RI. The *Nde*I–*Eco*RI fragment was cloned into the corresponding sites of pColdIII-NFH (Appendix A). The *Tri101* sequence, *N*-terminally fused with a FLAG-HA tag, was amplified by PCR from the resulting plasmid using primers FLAG-HA-Tri101F_NheI (5′-AAAAGCTAGCATGGACTACAAGGACGACGAT-3′) and FLAG-HA-Tri101R_XhoI (5′-AAAACTCGAGCTAACCAACGTACTGCGCATA-3′). After double digestion with *Nhe*I and *Xho*I, the PCR product was inserted between these sites downstream of a MMTV-LTR promoter in pMAM2BSD [25], yielding a DEX-inducible *Tri101* expression vector, pMAM2BSD_FH_Tri101.

### 3.5. Transfection of FM3A Cells and Selection of Transformants

The plasmid pMAM2BSD_FH_Tri101 or pMAM2BSD was transfected into FM3A cells using Lipofectamine^®^ 2000 reagent (Thermo Fisher Scientific, Waltham, MA, USA) following the manufacturer’s instructions. Plasmid DNA (5 µg) was diluted in 100 µL Opti-MEM medium without FBS. Five microliters of Lipofectamine^®^ 2000 was diluted in 100 µL of Opti-MEM medium and left for 5 min. Diluted plasmid DNA and Lipofectamine were mixed gently and left for 20 min and added to FM3A cells (7~10 × 10^5^/3 mL).

Transfected cells were incubated for one day before being transferred to N-medium containing 60 µg/mL of blasticidin S (Fujifilm Wako Pure Chemical Co., Osaka, Japan). During selection, the concentration of blasticidin S was increased to 250 µg/mL over one month. For the next selection step, the cells were transferred to B-medium containing 50 µM DEX and 0.1 µg/mL (0.4 µM) ITDol. The ITDol concentration was increased to 0.2 µg/mL (0.8 µM) over three weeks. The selected transformants (>100 cell lines) were cloned using limited dilution methods in B-medium containing 50 µM DEX and 0.04 µg/mL ITDol (0.16 µM). After limited dilution, both DEX and ITDol were removed from the medium. Among the survived transformants in this condition, the cell line whose crude enzyme showed the highest 3-*O*-acetyltransferase activity was chosen and designated as G3. Measurement of activities of prepared crude enzymes was performed as described below.

### 3.6. Preparation and Reaction of Crude Enzymes from Cultured Cells

Wild-type FM3A cells and the selected transformants were seeded (10 mL aliquots of cell suspension at 1 × 10^4^/mL). After a three-day incubation period, DEX (50 µM at final concentration) was added for the induction of *Tri101* expression in the transformed cells, and for comparison, in WT cells. In parallel, the carrier solvent (10% dimethylsulfoxide [DMSO]; 0.1% DMSO at a final concentration) was added to these cells. Once the cells became semi-confluent, these were harvested and washed twice in 2 mL PBS. The washed cells were suspended in 500 µL of 125 mM Tris-HCl buffer (pH 6.8) and sonicated. After centrifugation (13,000 rpm at 4 °C for 5 min), the supernatant was subjected to a protein quantitation assay (Pierce^TM^ BCA protein assay kit, Thermo Fisher Scientific). The resulting crude enzyme fraction was assayed for modification activity against C-3 of the trichothecene skeleton.

To initiate the reaction, 100 µL of the crude enzyme (with a protein concentration around 1.0 mg/mL) was mixed with 100 µL of substrate solution, which contained a 3-hydroxy (169 µM DON or 200 µM ITDol) or 3-acetyl (148 µM 3-ADON or 171 µM ITD) trichothecene in 125 mM Tris-HCl buffer (pH 6.5), respectively, with or without 1 mM acetyl CoA. We used rTRI101 from *Escherichia coli* (BL21 (DE3)-pColdIII-*Tri101*) (1 mg/mL) [30] as a positive control and boiled rTRI101 as a negative control. Each enzyme assay was performed in triplicate. The mixture was incubated for 3 h at 37 °C and the reaction was terminated by adding 200 µL of ethyl acetate. Trichothecenes were extracted with ethyl acetate three times, concentrated under N_2_, resuspended in the appropriate amount of ethanol, and subjected to TLC and HPLC analysis.

### 3.7. TLC and HPLC Analysis

The trichothecenes in each sample were detected roughly by TLC and the concentration of each trichothecene was precisely measured by HPLC. The samples dissolved in ethanol were applied to TLC plates (Merck, Darmstadt, Germany) and developed with ethyl acetate/toluene (3:1). Trichothecenes on TLC plates were visualized using the 4-(p-nitrobenzyl)pyridine-tetraethylenepentamine method [31].

For HPLC analysis, the samples dissolved in ethanol were filtered, and 10 µL was applied to HPLC (LC-2000 plus; Jasco, Tokyo, Japan) loaded with a C_18_ reverse-phase column (4.6 φ × 250 mm, PEGASIL ODS SP100). Trichothecenes were eluted with a mobile phase of acetonitrile and water, according to the following steps: for DON and 3-ADON, 20% acetonitrile for 6 min, a linear gradient of 20%–50% acetonitrile for 9 min, 50% acetonitrile held for 6 min, then 20% acetonitrile held for 10 min; for ITDol and ITD, 20% acetonitrile for 6 min, a linear gradient of 20%–100% acetonitrile for 16 min, 100% acetonitrile held for 3 min, then 20% acetonitrile held for 10 min. Deoxynivalenol and 3-ADON were detected at 254 nm, while ITDol and ITD were detected at 195 nm. The molar concentration of each trichothecene was calculated from the peak area, based on the previously obtained standard curve.

### 3.8. Toxicity Evaluation of Trichothecenes

In order to evaluate the effect of *Tri101* gene transfection into FM3A cells, we counted the number of live WT and G3 cells, to which ITDol had been added beforehand. First, the cells were seeded at 5 × 10^4^/mL in B-medium and were treated with 50 µM DEX. After one day of incubation, 1 mL of the cells were transferred to a 24-well plate and ITDol (0, 0.4, 0.8, 1.6, 4.0, or 8.0 µM ITDol in 0.1% DMSO at a final concentration) was added and incubated for 3, 5, or 7 days in a CO_2_ incubator at 37 °C. The cells were carefully harvested and centrifuged (300× *g*, 5 min, 25 °C). The precipitated cells were then tapped gently, and the appropriate amount of medium was added. The cell suspension was diluted twofold with 0.5% trypan blue (Nacalai Tesque) and only the live cells were counted.

For the cytotoxicity evaluation, colorimetric cell viability assays using WST-8 were performed. Each trichothecene was prepared in 50% DMSO as follows: for DON, 0, 0.0034, 0.010, 0.034, 0.10, 0.34, 1.0, or 2.0 mM (0, 0.001, 0.003, 0.01, 0.03, 0.1, 0.3, or 0.6 mg/mL); for 3-ADON, 0, 0.030, 0.089, 0.30, 0.89, 3.0, 8.9, or 30 mM (0, 0.01, 0.03, 0.1, 0.3, 1.0, 3.0, or 10.0 mg/mL); for ITDol, 0, 0.004, 0.008, 0.016, 0.040, 0.080, 0.16, or 0.40 mM (0, 0.001, 0.002, 0.004, 0.01, 0.02, 0.04, or 0.10 mg/mL); for ITD, 0, 0.0068, 0.014, 0.034, 0.068, 0.14, 0.34, 0.68, 1.4, 3.4, or 6.8 mM (0, 0.002, 0.004, 0.01, 0.02, 0.04, 0.1, 0.2, 0.4, 1.0, or 2.0 mg/mL). Wild-type or G3 cells were suspended at a cell density of 5 × 10^4^/mL in B-medium with or without 50 µM DEX and incubated for one day. Into each well of 96-well plates, 99 L of cell suspension was seeded and 1 µL of each trichothecene prepared above or 50% DMSO (vehicle) was added in triplicate. The cells were incubated for 48 h in a CO_2_ incubator. Into each well, 10 µL of Cell Counting Kit (CCK)-8 solution (Dojindo Molecular Technologies, Inc., Kumamoto, Japan) was added, and the microtiter plate was incubated for 3 h at 37 °C. OD_450_ was measured using a Multiskan^TM^ FC plate reader (Thermo Fisher Scientific), and growth inhibition (%) caused by each trichothecene was calculated.

### 3.9. Statistical Analysis

Statistical analysis was performed using Student’s *t*-test. Regarding the calculation and statistical analysis of IC_50_ values, log-logistic model was applied using R version 3.5.0 (R project for statistical computing).

## Figures and Tables

**Figure 1 toxins-11-00654-f001:**
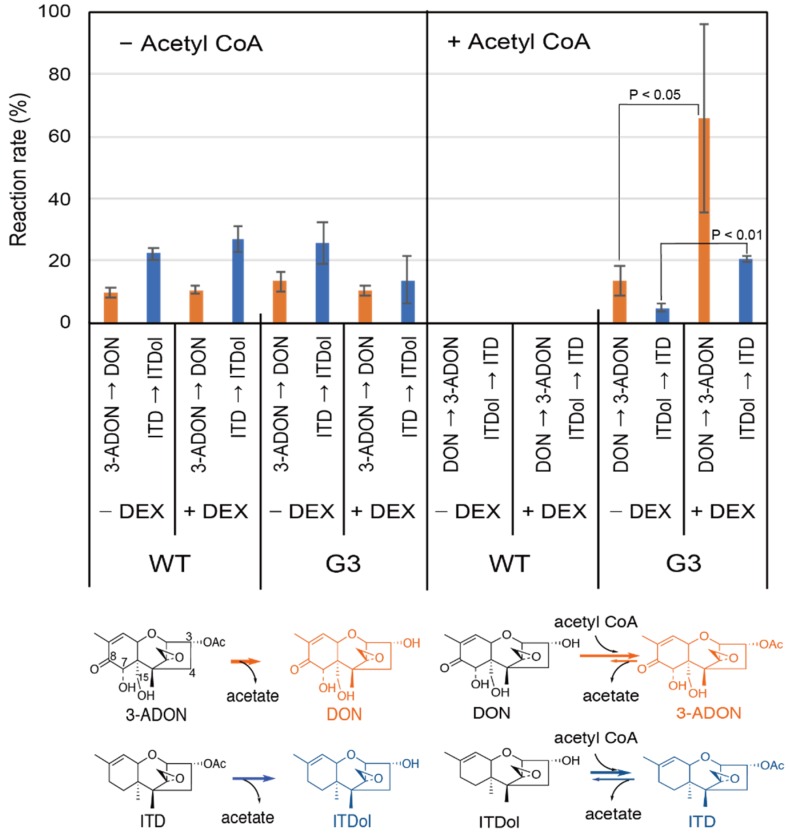
Acetylase and deacetylase activities of crude enzymes from FM3A cells. Wild-type (WT) and the transformant (G3) cells were pretreated with or without DEX (dexamethasone) and the crude enzyme was prepared from the harvested cells. The enzyme reaction was carried out in 200 µL reaction mixture, with (**right**) or without (**left**) 1 mM acetyl CoA, and 10 µg of a trichothecene as a substrate; 148 µM 3-ADON (3-acetyldeoxynivalenol), 171 µM ITD (isotrichodermin), 169 µM DON (deoxynivalenol), or 200 µM ITDol (isotrichodermol) at a final concentration. The trichothecene denoted at the root of the arrows represents the substrate, while the trichothecene at the tip of the arrow is the product. The percentage reaction rate (%) represents the initial molar ratio of the product over the added substrate per protein (1.0 mg/mL) in the reaction mixture. The values represent the average ± standard deviation (SD) (*n* = 3).

**Figure 2 toxins-11-00654-f002:**
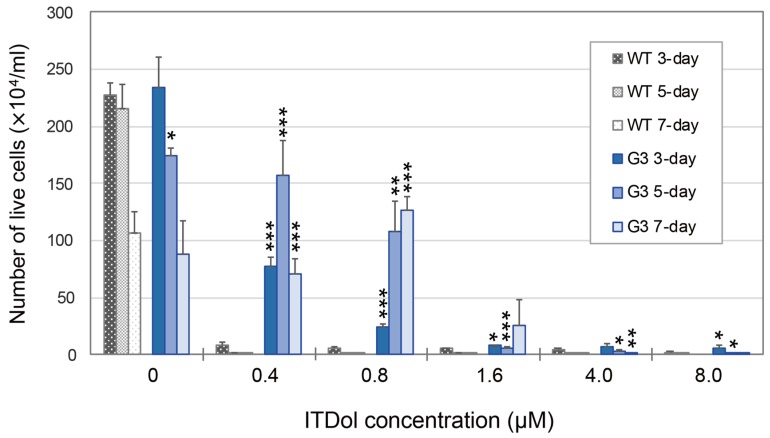
Growth of WT and the transformant G3 cells in B-medium containing ITDol. After pretreatment with DEX, the cells were seeded in 24-well plates, and ITDol solution was added to the cells. On day 3, 5, and 7, the cells were harvested and diluted with trypan blue solution, and the number of the live cells were counted. The values represent the average ± SD (*n* = 3). The numbers of live cells of G3 which showed statistical differences from those of their corresponding WT are marked with asterisks (* *p* < 0.05; ** *p* < 0.01; *** *p* < 0.001).

**Figure 3 toxins-11-00654-f003:**
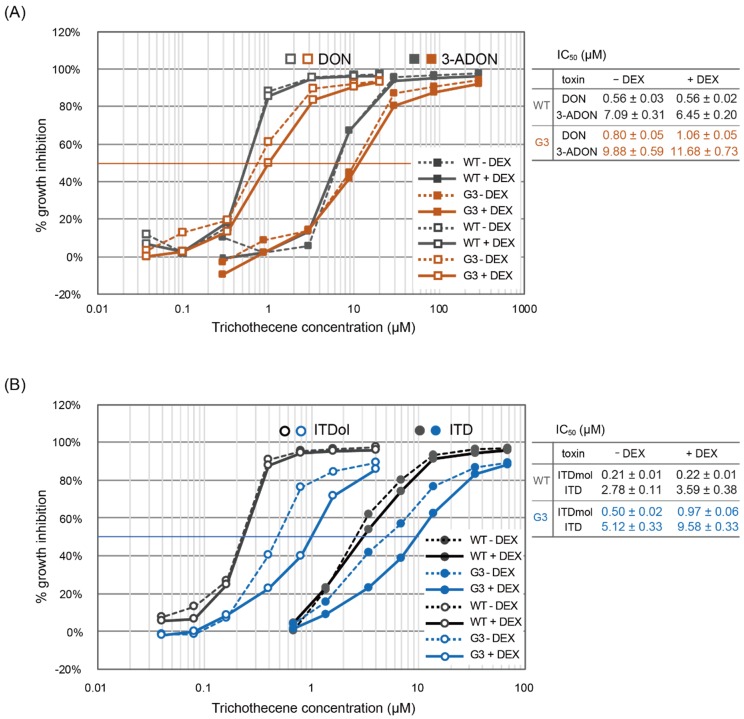
The dose–response cytotoxicity curves of the trichothecenes. (**A**) DON or 3-ADON and (**B**) ITDol or ITD was used as a toxin. Cytotoxicity assay of trichothecenes on FM3A cells was carried out using Cell Counting Kit 8 (CCK-8) reagent in 96-well plates. Cells were pretreated with or without 50 µM DEX. One microliter of a toxin or vehicle was added to 99 µL of cell culture, which was seeded (5 × 10^4^/mL) one day before. After two days of incubation with a toxin or vehicle, a WST assay was performed. Growth inhibition (%) was calculated as follows: 100 × {(OD_450_ of vehicle control − OD_450_ of background) − (OD_450_ of trichothecene added − OD_450_ of background)}/(OD_450_ of vehicle control − OD_450_ of background). The IC_50_ values represent the average ± standard error.

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
