# Peer review of "Reduced Toxicity of Trichothecenes, Isotrichodermol, and Deoxynivalenol, by Transgenic Expression of the Tri101 3-O-Acetyltransferase Gene in Cultured Mammalian FM3A Cells"

_toxins, 2019, doi:10.3390/toxins11110654_

Round 1

Reviewer 1 Report

Interesting and valuable work that expands our knowledge on the impact of trichothecene toxicity in mammalian cells. 

A few comments and suggestions:

Line 48.  

Trichothecenes are possible phytotoxins...

I would remove the word "possible" because it has been clearly shown using F.graminearum mutants with tri5 deletions impacts virulence in wheat (Jansen, et al. Infection patterns in barley and wheat spikes inoculated with wild-type and trichodiene synthase gene disrupted fusarium graminearum. Proceedings of the National Academy of Sciences of the United States of America 2005, 102, 16892-16897.).  Also, additional work shows trichothecenes induce ROS in plants and cause bleaching of vegetative tissues (ex. McLaughlin, et al. "A lipid transfer protein increases the glutathione content and enhances Arabidopsis resistance to a trichothecene mycotoxin." PloS one 10, no. 6 (2015): e0130204.)

Line 103:  Transfection should include an empty vector control to enable you to show any effect of the vector/backbone itself on gene expression in your target cells.

Line 139 Fig1. Legend.  Addition of mean comparison statistics would allow you to indicate significant differences in reaction rates in the test.  Similiar comment for Line 149 (Fig 2).

Line 158- What is the variance of the IC50 values?

Supplementary Table S1.  Add amount of time for incubation to the title would enhance readability.

Line 112 and Supplementary Figure S2A- regarding the amount of Tri101 enzyme produced and how much after a time course induction with DEX.  It is not clear if the resulting pMAM2BSD FH Tri101 plasmid contains the transgene Tri101 having the FLAG-tag, HA tag, 6 X His tag.  If so, it be valuable to check expression of the recombinant Tri101 enzyme using Western analysis.  I say this because in yeast, the length of time inducing the yeast promoter can be checked nicely via Western and that gives a better idea of how much pre-induction would be ideal prior to starting the experiment in the presence of trichothecenes.  For example, we find that at least a 6 hour induction time period is required to build up a certain amount of recombinant protein following the addition of galactose when using a GAL promoter.

Reviewer 2 Report

Article entitled “Reduced toxicity of trichothecene by transgenic expression of the Tri101 3-O-acetyltransferase gene in cultured mammalian cells” describes the effect of expression of Tri101 gene on mammalian cells on their ability to detoxify DON or ITDmol. The article is interesting and at the end the data support the hypothesis that acetylation at C3 results on a reduction of the citotoxicity of the resulting compounds.

However some important issues could be improved before publication:

- The problem of the existence of a strong C-3 deacetylase activity on the mammalian cells has not been totally overcome in this work.

- Authors could be more accurate in the description of data that allowed them to conclude that the toxicity of 3-OH trichothecenes on mammal cells was overestimated in previous works.

Some additional points can be also improved at the Introduction:

Lines 38.- There are other many non-Fusaria genera producting trichothecenes, please be more accurate and include a reference (e.g. Proctor et al., 2018; doi: 10.1371/journal.ppat.1006046).

General comment to the introduction.- It is not clear for this reviewer what is the importance of transfecting cells with Tri101 gene to evaluate toxicity of the 3-OH or 3-acetyl derivatives, once authors can not confirm.

Reviewer 3 Report

Lines 31-32: The classification of trichothecenes in two groups namely t-type and d-type based on the biosynthetic pathways is right, but authors should also mention the chemical classification in Types (A, B, C, and D) naming which Type is biosynthetized by each pathway.

One of the objectives of the work is to improve the accuracy of the evaluation of the toxicity of 3-acetyl trichothecenes, but could you define the practical applications of the transfection into mammalian cultured cells to improve their resistance to 3-hydroxytrichothecenes?

Reviewer 4 Report

I believe this manuscript is interesting and worthy of publication after minor revisions.  I recommend the authors add a paragraph in the discussion section.  They should suggest a practical application of their findings.  Even though the creation of transgenic livestock is controversial in many nations, the data of this manuscript indicates that transgenic expression of Tri101 in the digestive systems of livestock could possibly mitigate trichothecenes present in ingested grain.

I suggest figure 1 incorporate statistical analysis, especially to determine if DEX addition results in a statistically significant increase in acetylation rate. 

In addition, I think the IC50 values of figure 3 should include a statistical analysis. 

I have a number of minor editing suggestions:

Line 49.  Use enables instead of contributes

Line 63-64.  Suggestion: "there is a conflicting report of 3-acetyl"

Line 98.  Suggest "trichothecenes in a cell culture"

Line 162.  Suggest "in the formation of more ITDol"

Line 197.  Suggest "and that their cytotoxicities were overestimated in other studies."

Line 298.  I am not sure what the symbol following 4.6 means. 

Round 2

Reviewer 2 Report

Authors addressed correctly the concerns raised by this reviewers. Thus, even when some issues have not been corrected, authors explain these items properly. Thus, I think the article can be accepted in the present form for publication.